# Consequences of Transplacental Transmission of the SARS-CoV-2 Virus: A Single-Center Experience

**DOI:** 10.3390/children9071020

**Published:** 2022-07-08

**Authors:** Ivona Djordjevic, Ana Kostic, Ivana Budic, Nikola Vacic, Zlatan Elek, Strahinja Konstantinovic

**Affiliations:** 1Pediatric Surgery Clinic, University Clinical Center Nis, Dr Zorana Djindjica Blvd. 48, 18000 Nis, Serbia; veljkodjor@gmail.com (A.K.); vacic1410@gmail.com (N.V.); strahinja0403@hotmail.com (S.K.); 2Faculty of Medicine, University of Nis, Blvd Zoran Djindjic 81, 18108 Nis, Serbia; ivona.djordjevic@medfak.ni.ac.rs; 3Clinic for Anesthesiology and Intensive Care, University Clinical Center Nis, Dr Zorana Djindjica Blvd. 48, 18000 Nis, Serbia; 4Clinical Hospital Center Kosovska Mitrovica, Anri Dinan Street 10, 38220 Kosovska Mitrovica, Serbia; drzelek@gmail.com; 5Faculty of Medicine, University of Pristina, Filip Visnjic Street bb, 38220 Kosovska Mitrovica, Serbia

**Keywords:** SARS-CoV-2, transplacental transmission, SARS-CoV-2 coagulopathy, neonate

## Abstract

The issues of vertical viral transmission from mother to fetus and the potential complications caused by SARS-CoV-2 coagulopathy are still unclear. There are few literature data about the vertical transmission of SARS-CoV-2 and health outcomes in neonates born to mothers with symptomatic or asymptomatic coronavirus disease, with the existing data based on small sample sizes. This case series study consists of two newborn children (one pre-term and one term) who were born to SARS-CoV-2-positive mothers and admitted to the neonatal intensive care unit a few hours after birth. One child had cyanotic changes that affected the entire left leg and the left forearm, with multiple livid changes on the front of the chest and abdomen, the right upper arm, right thigh, neck, and face, and one child had an altered umbilical cord. The first child was treated conservatively, and the second child was treated surgically.

## 1. Introduction

The SARS-CoV-2 pandemic has had catastrophic consequences globally, especially in underdeveloped and developing countries. However, the issues of vertical viral transmission from mother to fetus, and especially the potential complications caused by SARS-CoV-2 coagulopathy, remain unclear. According to the data, patients with the following conditions have an increased risk of severe disease: oncological patients, patients with chronic diseases, immunocompromised patients, and pregnant women [1]. The literature describes different pregnancy outcomes in SARS-CoV-2-positive mothers, namely abortion [2], pre-term birth [3], pre-eclampsia [4], severe respiratory tract infection in mothers [5], mortality [6], neonatal complications and perinatal death [7,8,9], and vertical viral transmission from mother to fetus. The placenta is a highly specific organ that develops during pregnancy, whose role is to form and maintain an optimal environment for normal fetal development. A wide range of viral infections are associated with characteristic histopathological changes in the placenta [10]. The analysis of the placental tissue of mothers with SARS-CoV-2 showed the existence of avascular chorionic villi [11]. However, the analysis of the placental tissue of SARS-CoV-2-positive mothers is limited to only a few cases in the literature describing histiocytic intervillositis with spike protein in the syncytiotrophoblast [12,13]. Moreover, histopathological examination of SARS-CoV-2-positive pregnant women demonstrates fetal vascular malperfusion and phlebopathy with the detection of blood clots [12].

## 2. Materials and Methods

The study was designed as a retrospective case study series. Both patients were newborn children (one pre-term and one term) who were admitted a few hours after birth and born to SARS-CoV-2-positive mothers. The infants were admitted to the neonatal intensive care unit for further diagnostic examinations and treatment. Diagnoses were established using biochemical analyses, a complete blood count (CBC), coagulation factors screening, bleeding and coagulation time, D-dimer value, a nasopharyngeal swab for SARS-CoV-2, and a Doppler ultrasonography examination.

## 3. Case Presentations

We present a case series of newborn children presenting the consequences of transplacental transmission of SARS-CoV-2. This case series included two children who were admitted to the neonatal intensive care unit in the Pediatric Surgery Clinic, University Clinical Centre of Nis, Serbia, immediately after birth. One of them was treated conservatively and the other was treated surgically.

### 3.1. Case Report 1

A previously fully vaccinated 27-year-old pregnant woman with no comorbidities (gravida 1, para 0) was admitted to the gynecology clinic due to frequent contractions in the 35^th^ week of gestation. The pregnancy was monitored regularly, and the course of the pregnancy was orderly. The pregnant woman, on admission to the hospital, did not present any problems and was afebrile. A routine test (nasopharyngeal swab) for SARS-CoV-2 showed a positive result, and the pregnant woman was isolated. Contractions were frequent, and the pregnant woman was prepared for childbirth. The delivery ended with a cesarean section, without any intraoperative or postoperative complications for the pregnant woman.

At birth, the boy weighed 2850 g and measured 50 cm in length; his Apgar score was 7/9. Immediately after birth, cyanotic changes affected the entire left leg and the left forearm, with multiple livid changes on the front of the chest and abdomen, the right upper arm, right thigh, neck, and face (Figure 1).

The child was intubated and transferred to the neonatal intensive care unit for further diagnosis and treatment. Initial biochemical analyses, CBC, coagulation factor screening, bleeding and coagulation time, D-dimer, and a nasopharyngeal swab for SARS-CoV-2 were performed. The swab was positive and there was a high titer of SARS-CoV-2 IgM antibodies, which was interpreted as indicating the intrauterine transmission of the virus from mother to child during pregnancy. The D-dimer was five times higher than normal, with a C-reactive protein of 125 mL/L and leucocytes up to 22 × 109/L. The left leg was cold, with hyporeflexia and filiform pulsations (Figure 2). The left forearm was also cold and livid with a weak, almost undetectable pulse in the radial artery.

A Doppler ultrasonography examination of all the extremities was performed. The flow through the right leg and forearm was irregular. A flow through the left radial artery and ulnar artery was detected, although it was significantly weakened due to thrombotic masses in the arteries. A scarce flow was also detected through the arteries of the left leg due to thrombotic changes predominantly in the left femoral artery. There were no visible signs of compartment syndrome. Due to the still-present flows, the possibility of amputation was postponed and conservative treatment with unfractionated heparin was started. Gentamycin and ampicillin were empirically included at the beginning. Additional biochemical analyses, sampling materials for urine and blood culture, skin swabs, and further diagnostic tests followed almost every day. After five days of therapy, the local findings improved, and the extremity color recovered. A more intense pulse wave was registered, but the left leg still had hyporeflexia. Therapy was continued for the next seven days until complete regression of liquidity. The leucocytes and C-reactive protein were within regular range. The D-dimer had a decreasing manner. The patient was extubated a day later, and a physiatrist and a neurologist were involved in the patient’s treatment process. The patient was discharged home on day 35 without further complications, with mild hyporeflexia and hypotonia of the left leg.

### 3.2. Case Report 2

A 33-year-old pregnant woman was hospitalized in the gynecology clinic due to frequent contractions in the 38th week of gestation and a fever up to 37.5 °C, with mild myalgia and arthralgia. The woman was previously healthy and fully vaccinated. The course of the pregnancy was orderly, and the pregnant woman went for regular check-ups. No deviation from typical morphology in the fetus was noticed. Double and triple tests were negative. A test for SARS-CoV-2 was positive, and the soon-to-be mother was isolated. The birth was completed naturally in the next 17 h, without any complications.

A boy was born who was 52 cm in length, with a body weight of 3250 g and an Apgar score of 7/9. The child cried immediately after birth and had a normal turgor of the skin, rhythmic heart action, and clear tones. The findings in the lungs were normal. The existence of an altered umbilical cord in the umbilical cord area was recorded.

Thrombotic masses filling the umbilical vein could be seen through the layers of the Wharton’s jelly (Figure 3). Soon after the initial resuscitation, the child was transferred to the neonatal intensive care unit for further treatment.

Standard biochemical analyses, CBC, an abdominal ultrasound, coagulation factor screening, bleeding and coagulation time, a thrombophilia test, D-dimer, and a nasopharyngeal swab for SARS-CoV-2 were performed. The levels of the leucocytes and C-reactive protein were regular, with an elevated D-dimer up to 1200 ng/mL. The findings did not deviate from expected values, and thrombophilia was excluded. As expected, the test for SARS-CoV-2 was positive with a high titer of IgM antibody (transplacental transmission of SARS-CoV-2). An ultrasound of umbilical cord thrombosis was verified as a hyperechoic echo within the umbilical vein, with a very discrete edge flow through the venous vessel at the site of placental insertion of the umbilical cord. The entire umbilical cord presented as a highly edematous, ectatic, tortuous, and voluminous structure. The other (intra-abdominal) part of the umbilical vein was without varicosity and without thrombus. Findings on the abdominal organs were normal. The liver did not have focal lesions. The flow through the portal vein was normal.

Due to the isolated extra-abdominal localization of the thrombus in the umbilical vein, a decision was made to remove the thrombus surgically by opening the umbilical vein in the umbilical cord, and then to ligate the umbilical cord above the skin insertion. According to the recommendations of a clinical pharmacologist, unfractionated heparin and ampicillin were included in the therapy.

Control Doppler ultrasonography examinations were performed 3, 7, and 10 days after birth, and did not show the existence of thrombotic masses in the intra-abdominal part of the umbilical vein. Blood tests and biochemical analyses were within normal ranges, D-dimer was decreasing, and the patient was discharged home without further complications on day 12.

Literature data about the vertical transmission of SARS-CoV-2 and health outcomes in neonates born to mothers with symptomatic or asymptomatic coronavirus disease are rare, based on small sample sizes, and come primarily from China, whose results might not generalize to all populations. According to studies, the incidence of vertical transmission of SARS-CoV-2 infection is currently low, with low rates of testing-based vertical or perinatal transmission and no clinical evidence of neonatal SARS-CoV-2 infection [13]. Neonates born to mothers with severe or critical illness were mostly born at an earlier gestational age.

Very little is currently known about the effects of SARS-CoV-2 on the human placenta and neonate [14]. Human SARS has been vertically transmitted and, in some cases, has shown fetal thrombotic vasculopathy (fetal vascular malperfusion) [11]. Early evidence did not demonstrate the vertical transmission of SARS-CoV-2 in small cohorts of human patients [5,6]. However, the identification of IgM antibodies to SARS-CoV-2 in a single neonate, who also had elevated cytokines, suggests that vertical transmission is possible, even if it is uncommon [15]. The SARS-CoV-2 infection has been associated with hypercoagulability and the development of ischemic changes, D-dimer elevation, and, in some patients, disseminated intravascular coagulopathy [16,17]. In one study, a few patients showed fetal vascular malperfusion, which was potentially related to aberrant cord insertion [18]. However, in most cases, there was no gross umbilical cord abnormality known to be associated with fetal vascular malperfusion. This suggests that, in some cases, fetal SARS-CoV-2 infection might be associated with a propensity for thrombosis in the fetal circulation.

The prevention of coronavirus-induced illness in pregnant women is crucial because pregnancy poses a risk of a severe form of the disease. The UK Medicines Agency analyzed available data from England, and, based on their report, coronavirus vaccines were deemed safe during pregnancy. The rate of premature births was 6.51% in vaccinated women and 5.99% in unvaccinated women [19]. The hypercoagulable condition in the SARS-CoV-2 virus is still subject to scrutiny, with several different mechanisms trying to explain it. The results of in vitro testing for the SARS-CoV-1 virus (which has a similar binding mechanism to ACE2 receptors as SARS-CoV-2) showed increased gene expression for fibroblast growth factor (FGF), fibrinogen gamma chain (FGG), and serin protease, which can lead to a procoagulable condition by activating several coagulation factors (II, III, and X). McGonagle claimed that the SARS-CoV-2 virus, by binding to ACE2 receptors, presented predominately on type II pneumocytes, leading to an inflammatory cascade that causes hypercoagulation in the lungs and increased expression of tissue factors on the endothelial cells, macrophages, and neutrophils, thus activating the coagulation cascade [20]. Long Ji and associates claimed that the hypercoagulable condition may be due to antiphospholipid syndrome, with the detection of elevated values of antiphospholipid antibodies in SARS-CoV-2-positive patients [21], while Campbell emphasized the importance of activating complements in patients with severe forms of the infection [22]. Accordingly, additional evidence is needed to determine whether fetuses may be at theoretical risk of intrauterine thrombotic events, or any other events induced by maternal infection with SARS-CoV-2.

Literature data emphasize the impacts of SARS-CoV-2 infection on blood viscosity and the aggregation of red blood cells. In a study of 172 patients hospitalized in the COVID Department of Internal Medicine in Lyon, France, Nader and associates analyzed routine hematological and biochemical parameters, blood viscosity, deformity, and the aggregation of red blood cells on the first day of hospitalization. Patients with SARS-CoV-2 infection, despite showing reduced hematocrit, showed increased blood viscosity and increased aggregation of the erythrocytes. The authors of the study also observed a connection between more significant aggregation and better coagulation in patients with pulmonary lesions and in patients treated with oxygen supplementation [23]. All of this favors a multisystem influence of SARS–CoV-2 on the human body.

Thromboembolic complications in newborns carry a high risk of morbidity and mortality. Thrombosis, depending on localization, can lead to the damage of all organs, including the lungs, heart, intestines, and brain, and to damage of the extremities. Before the SARS-CoV-2 pandemic, in 2013, Szvetka and associates published an article about neonatal thrombosis. They believed that the embolus originated in the placenta and later passed through the umbilical vein, then traveled through the liver and the lower caval vein to the right and left heart chamber, and then continued on the path to the right axillary artery and carotid. However, there were no signs of abnormalities in the umbilical cord [24]. From the abovementioned data on the impact of SARS-CoV-2 infection on the hypercoagulable condition, we can assume that the acute ischemia of the extremities in one of our cases can be explained in this way. On the other hand, Perveen and associates suggest another possible cause of ischemia of the extremities in newborns. They pointed out the important role of compartment syndrome, characterized by increased hydrostatic pressure in the muscles, which compromises microcirculation and causes tissue ischemia, decreasing capillary and, later, venous and arterial circulation. The causes of this syndrome in neonates can be divided into external ones, such as oligohydramnios and umbilical cord compression, and internal ones, such as arterial embolism or neonatal hypercoagulation. The authors of the study claimed that neonatal ischemia can manifest at varying degrees but, interestingly, without rhabdomyolysis. One of the causes of oligohydramnios can be placenta size. Placenta enlargement, which reduces the amount of fluid, restricts hand movement, leads to compression and consequential ischemia, and can result in the onset of compartment syndrome [18]. In our case, which concerned ischemia of the leg, compartment syndrome was not detected, and we concluded that the ischemia was the result of thromboembolic complications and the thrombosis of the umbilical vein.

Data concerning the relationship between the SARS-CoV-2 virus and its impact on pregnancy are developing very quickly. The question of whether the level of placental perfusion disorder is due to a hypercoagulable condition, or the result of inflammation of the villi as part of the immune response to the presence of the virus, requires further study.

## 4. Conclusions

This article describes complications resulting from the vertical transmission of the SARS-CoV-2 virus from a SARS-CoV-2-positive mother to her fetus, highlighting the significant impact of SARS-CoV-2 infection on neonatal morbidity. The exact pathophysiological mechanism of these complications is at the molecular level and is still subject to scrutiny. Since SARS-CoV-2 infection carries a high risk of morbidity and mortality in mothers and neonates, it is essential to take all measures to prevent SARS-CoV-2 infection and thus prevent these complications. 

Ethical review and approval were waived because this study included two isolated clinical case reports for which we obtained the parents’ written consent to enroll their children in the study.

## Figures and Tables

**Figure 1 children-09-01020-f001:**
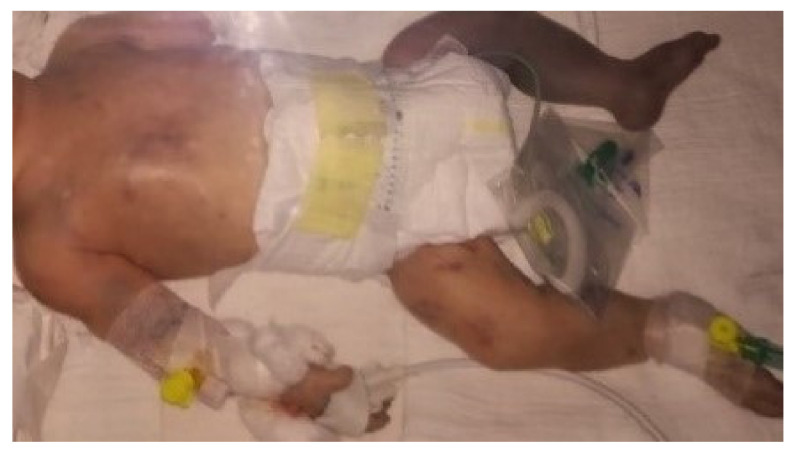
Newborn baby with cyanotic changes affecting the entire left leg and the left forearm, with multiple livid changes on the front of the chest and abdomen, the right upper arm, right thigh, neck, and face.

**Figure 2 children-09-01020-f002:**
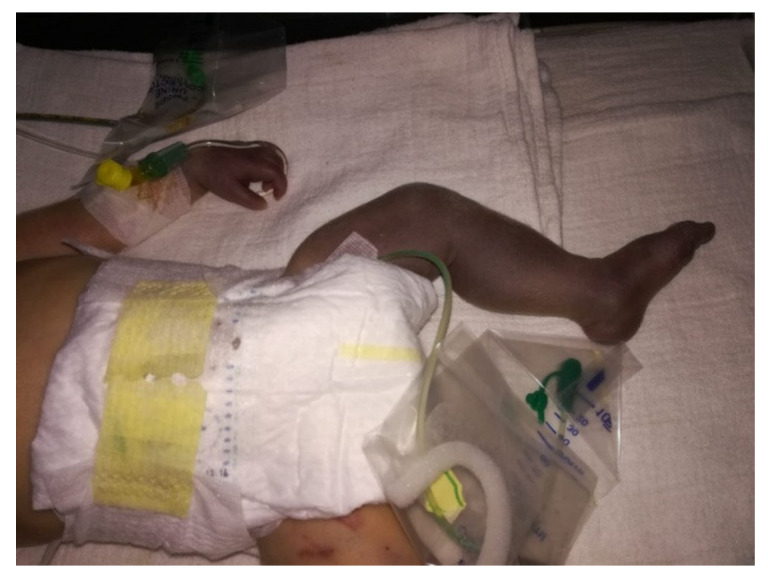
The most severe ischemic changes on the whole left leg, left hand, and forearm.

**Figure 3 children-09-01020-f003:**
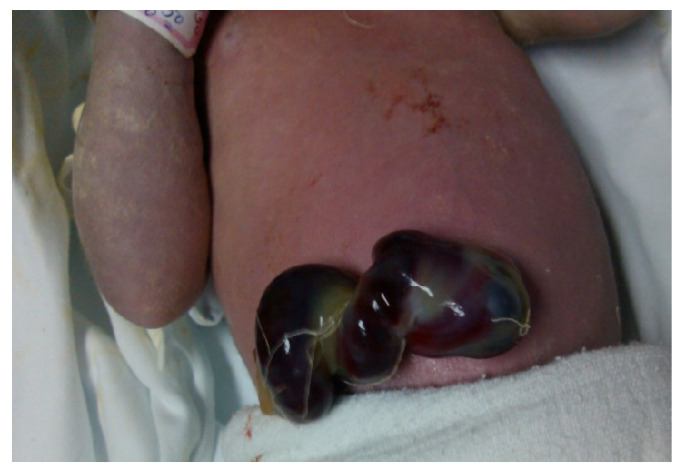
Newborn baby on his first day of life. The existence of an altered umbilical cord in the umbilical cord area (umbilical cord thrombosis) was recorded. The mother was positive for SARS-CoV-2 infection.

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
