# Peer review of "Consequences of Transplacental Transmission of the SARS-CoV-2 Virus: A Single-Center Experience"

_children, 2022, doi:10.3390/children9071020_

Round 1

Reviewer 1 Report

In the present study, the authors present their experience regarding the possible vertical transmission of SARS-CoV 2, in relation to two cases in which the neonates suffered vascular and thrombo-embolic complications.

I find the article interesting, as it adds to the body of evidence in this regard. However, I believe that there are some questions that should be resolved before it can be accepted.

1. In the exposition of the cases, there is a lack of information regarding the mother’s health. It seems important to highlight whether the patients had any underlying pathology, whether they were vaccinated (since reference is made to this aspect in the discussion), and, above all, whether they received prophylactic treatment with enoxaparin after diagnosis and at what dose.

2. Regarding “Figure 1”, I consider it unnecessary to show the entire body of the neonate, since the findings appear in the lower regions, so I propose cropping the image from the neck of the neonate upwards.

3. In case 1, are there blood cultures that rule out infectious etiology as a cause of cyanosis?

4. Line 163: missing reference.

5. The conclusions section is unusually long, and seems more like a summary of the article. The authors should highlight one or two relevant aspects regarding the cases presented as a conclusion.

6. In the "Institutional Review Board Statement", they state the following: "Ethical review and approval were waived for this study, due to rare and unexpected complications of Covid infection in children". I do not believe that this is a reason for not requesting the approval of the ethics committee. Rather, the justification would be that it is an isolated clinical case report, and the publication of which was consented to by each of the patients involved.

In conclusion, I consider that the authors present two interesting and detailed cases, however, it is necessary to resolve these issues before publication.

Author Response

Dear Editor and Reviewers,

We would like to thank you for the quick review process and the helpful reviewers’ suggestions that we have adopted and that we hope have improved our manuscript.

We would like to thank all the reviewers for their dedicated time to make our work clearer and more acceptable for publication.

Below are responses to reviewers ’comments and suggestions.

Reviewer 1

Comments and answers

  1. In the exposition of the cases, there is a lack of information regarding the mother’s health. It seems important to highlight whether the patients had any underlying pathology, whether they were vaccinated (since reference is made to this aspect in the discussion), and, above all, whether they received prophylactic treatment with enoxaparin after diagnosis and at what dose.

Both mothers were fully vaccinated, previously healthy women with no history of comorbidities.

In the postoperative period, they received the prophylactic dose of enoxaparin (40 mg per day).

  1. Regarding “Figure 1”, I consider it unnecessary to show the entire body of the neonate, since the findings appear in the lower regions, so I propose cropping the image from the neck of the neonate upwards.

Thank you very much for this comment. The image is cropped as you kindly suggested.

  1. In case 1, are there blood cultures that rule out infectious etiology as a cause of cyanosis?

The initial swab of the throat, nose, skin, and umbilicus is part of the standard protocol for each newborn baby admitted to the NICU. Blood culture for this patient was done on day 2 according to the neonatologist's recommendation. The blood culture and all swabs were negative.

  1. Line 163: missing reference.

Many thanks for the suggestion. A reference is added.

  1. The conclusions section is unusually long, and seems more like a summary of the article. The authors should highlight one or two relevant aspects regarding the cases presented as a conclusion.

We corrected conclusion according to your recommendations.

  1. In the "Institutional Review Board Statement", they state the following: "Ethical review and approval were waived for this study, due to rare and unexpected complications of Covid infection in children". I do not believe that this is a reason for not requesting the approval of the ethics committee. Rather, the justification would be that it is an isolated clinical case report, and the publication of which was consented to by each of the patients involved.

You are absolutely right and thanks for the comment.

Because there were two isolated clinical case reports for whom we needed the parents' written consent in order to enroll them in the study, ethical review and approval were waived.

Reviewer 2 Report

The manuscript ID: children-1797959 entitled “Consequences of transplacental transmission of SARS-CoV2 2 virus - a single-center experience” reports SARS-CoV transplacental transmission in 2 newborns and their associated consequences. The study is really interesting but I have some concerns/comments that can help to improve it.

- The major weakness is about how or based on what the authors conclude that the clinical signs observed are due to the SARS-CoV-2 infection and not from another cause. These observations where never seen in other newborns not infected from SARS-CoV-2?

- The authors pointed out that one of the debilities of the data previously published is that are from small sample size studies, and they only describe two cases herein, which at least is somewhat controversial.

- I have some concerns about the ethical statement, since it may be necessary an ethical approval. In addition, what about the newborns consent? The authors state that the consent was obtained from all the subjects but the newborns can not do this, so if the parents signed the consent in name of them it should be clearly mentioned.

- There are several typo and/or grammatical errors.

Author Response

Dear Editor and Reviewers,

We would like to thank you for the quick review process and the helpful reviewers’ suggestions that we have adopted and that we hope have improved our manuscript.

We would like to thank all the reviewers for their dedicated time to make our work clearer and more acceptable for publication.

Below are responses to reviewers ’comments and suggestions.

Reviewer 2

Comments and answers

  1. The major weakness is about how or based on what the authors conclude that the clinical signs observed are due to the SARS-CoV-2 infection and not from another cause. These observations where never seen in other newborns not infected from SARS-CoV-2?

Given that thromboses in neonates are extremely rare and almost always accompany thrombophilia-type diseases or severe comorbidities in mothers who are excluded, and that the positive SARS-Cov-2 in the mother and the child and the positive IgM in the child have been proven, there is no other explanation than that it is a consequence of SARS-Cov-2 infections.

  1. The authors pointed out that one of the debilities of the data previously published is that are from small sample size studies, and they only describe two cases herein, which at least is somewhat controversial.

Many thanks for the comment.

This manuscript is a contribution to the previous publications that also reported a small number of patients due to an extremely rare condition.

  1. I have some concerns about the ethical statement, since it may be necessary an ethical approval. In addition, what about the newborns consent? The authors state that the consent was obtained from all the subjects but the newborns can not do this, so if the parents signed the consent in name of them it should be clearly mentioned.

Thank you so much for the suggestion. We obtained parental consent to publish the cases. It's just a clumsily used terminology that we've corrected.

In the supplementory file, we have provided signed written consent from parents who allow the inclusion of newborns in the study.

  1. There are several typo and/or grammatical errors.

A professional English translator edited the entire text.